

**Role of needle surface waxes in dynamic exchange of**
**mono- and sesquiterpenes**
**J. Joensuu[1], N. Altimir[1], H. Hakola[2], M. Rostás[3], M. Raivonen[4], M. Vestenius[2], H.**
**Aaltonen[2], M. Riederer[5] and J. Bäck[1,]**
[1]{Department of Forest Sciences, University of Helsinki, Finland}
[2]{Finnish Meteorological Institute, Helsinki, Finland}
[3]{Bio-Protection Research Centre, Lincoln University, Christchurch, New Zealand}
[4]{Division of Atmospheric Sciences, University of Helsinki, Finland}
[5]{Julius-von-Sachs-Institut für Biowissenschaften, University of Würzburg, Germany}
Correspondence to: J. Joensuu (johanna.joensuu@helsinki.fi)
**Abstract**
Biogenic volatile organic compounds (BVOCs) produced by plants have a major role in
atmospheric chemistry. The different physicochemical properties of BVOCs affect their
transport within and out of the plant as well as their reactions along the way. Some of these
compounds may accumulate in or on the waxy surface layer of conifer needles and participate
in chemical reactions on or near the foliage surface. The aim of this work was to determine
whether terpenes, a key category of BVOCs produced by trees, can be found on the
epicuticles of Scots pine (*Pinus sylvestris* L.) and, if so, how they compare with the terpenes
found in shoot emissions of the same tree. We measured shoot-level emissions of pine
seedlings at a remote outdoor location in Central Finland and subsequently analysed the
needle surface waxes for the same compounds. Both emissions and wax extracts were clearly
dominated by monoterpenes, but the proportion of sesquiterpenes was higher in the wax
extracts. There were also differences in the terpene spectra of the emissions and the wax
extracts. The results, therefore, support the existence of BVOC associated to the epicuticular
waxes. We briefly discuss the different pathways for terpenes to reach the needle surfaces and
the implications for air chemistry.



# 1 Introduction

At the border of the atmosphere and Earth's ecosystems, the living layer of vegetation is an active player interacting with its surroundings in multiple ways. Plants absorb, transmit and produce compounds like water, oxygen and carbon, as well as a myriad of more complex molecules such as volatile organic compounds (VOCs). In addition to this biological activity, plant surfaces provide area for adsorption, desorption and chemical reactions. These phenomena are affected by both environmental conditions and the structure (species, canopy layers etc.) of the vegetation – in turn shaping itself in response to the environment it grows in. The result of these interactions is an extremely complex and dynamic network of simultaneous processes.

Biogenic VOCs (BVOCs) produced by plants have a major role in atmospheric chemistry. They affect the formation and destruction of ozone in the troposphere and participate in aerosol formation processes (e.g. Kulmala et al., 2004, Tunved et al., 2006). Despite considerable progress in recent years, aerosol-related processes are a major source of uncertainty in climate estimates (IPCC 2014). Biogenic VOC emissions dominate over those of anthropogenic origin both globally (Guenther et al., 1995) and in the sparsely populated regions of Northern Europe, especially in the summertime (Simpson et al., 1999, Lindfors et al., 2000).

Terpenes (monoterpenes ($C_{10}H_{16}$) and sesquiterpenes ($C_{15}H_{24}$)) represent a reactive subgroup of BVOCs that are produced in different plant tissues and during various physiological processes (e.g. Loreto and Schnitzler, 2010). Plants are known to use these compounds in their interactions with insects and other plants, and they may help the plant to adapt to abiotic stress (see Holopainen and Gershenzon, 2010 for a review). BVOC emissions in the Eurasian taiga are dominated by monoterpenes (Guenther et al., 1995, Tarvainen et al., 2007, Rinne et al., 2009), but boreal forest trees also produce significant amounts of e.g. sesquiterpenes (Hakola et al., 2006, Holzke et al., 2006, Ruuskanen et al., 2007), which are generally more reactive than monoterpenes (Atkinson and Arey, 2003, Appendix A). Many terpenes are produced constitutively, but synthesis can also be induced by biotic and abiotic stresses such as herbivory or heat (Holopainen and Gershenzon, 2010, Loreto and Schnitzler, 2010). Plants store terpenes either in specialised storage structures like the resin canals of conifers on nonspecifically in the mesophyll tissue (Niinemets et al., 2004).





On their way from the plant interior to the atmosphere, the terpenes, mostly rather lipophilic
in nature (Niinemets and Reichstein, 2003, Appendix A), must first cross the lipophilic cell
membranes and then the hydrophilic apoplast before evaporating into the air spaces inside the
leaf. It was long assumed that this transfer happens purely by diffusion, but new evidence
suggests active transport out of the cells (Widhalm et al., 2015). Finally, emission into the
atmosphere occurs first by gas-phase diffusion through the stomata and the leaf boundary
layer, where the conditions are significantly affected by the leaf (Schuepp, 1993), and then by
turbulent transport. The driving force of diffusion is the concentration gradient between the
leaf interior and the atmosphere. The leaf cuticle is generally considered an effective barrier
for plant-produced volatiles, preventing direct emission (Niinemets and Reichstein, 2003).
The different physicochemical properties of terpenes affect their transport within and out of
the needle as well as their reactions along the way (Atkinson and Arey, 2003, Niinemets and
Reichstein, 2003, Appendix A). For example volatility (described by Henry's law constant H;
Pa m$^3$ mol-1) and partitioning between the lipid and aqueous phases (octanol-water partition
coefficient $K_{OW}$) vary between compounds, as do reaction rates with oxidants such as $O_3$.
Terpenes participate in many chemical reactions at and near the needle surfaces. For example,
terpenes can protect the plant from oxidative stressors such as ozone ($O_3$) by reacting with it
before it reaches the sensitive tissues inside the leaves (Loreto and Schnitzler, 2010). BVOC
reactions are known to be a major factor in non-stomatal $O_3$ deposition in forests (Goldstein et
al., 2004, Bouvier-Brown et al., 2009). The terpene-$O_3$ reactions can occur in the atmosphere
after terpene emission, but they can also take place in the leaf boundary layer, in the air spaces
or aqueous phase inside the leaf – or on the leaf surface (Altimir et al., 2006). In addition to
gas-phase reactions, heterogeneous reactions are known to play a key role in BVOC
chemistry (Shen et al., 2013). It has been suggested that some of the BVOCs produced by
foliage could be attached to the epicuticular waxes (Sabljic et al., 1990, Welke et al., 1998),
providing additional protection against oxidants, but scientific knowledge on this issue is
currently very limited. At least in theory BVOCs also affect the formation of water films on
leaf surfaces (Rudich et al., 2000, Sumner et al., 2004), thereby enhancing $O_3$ deposition
mediated by surface wetness.
The surfaces of conifer needles are both complex and dynamic in nature. As they grow,
needles are covered with a waxy layer secreted by the epicuticular cells (Fig. 1). This layer is
lipophilic and hydrocarbons are known to be taken up in it (Binnie et al., 2002, Brown et al.,





1998, Welke et al., 1998). With time and weathering, the surfaces undergo chemical and
structural changes (Barnes and Brown, 1990, Huttunen and Laine, 1983). Irregularities in the
surface provide sites for water adsorption (Rudich et al., 2000). As a result, the originally
water-repellent surface becomes more wettable as it wears down. Compounds accumulating
on the surface change the characteristics of both the surface and the water film that forms on it
(Neinhuis and Barthlott, 1997, Burkhardt and Eiden, 1994). Such water films are ubiquitous
when the ambient relative humidity is above 70 % – a common condition in boreal areas –
and can even extend through the stomata, creating a pathway for water-soluble compounds
between the leaf inside and the surface (Burkhardt et al., 2012).
Thus it is plausible that plant-derived terpenes with varying chemical properties could
accumulate on foliage surfaces in amounts and proportions difficult to predict and participate
in reactions with other compounds. Because of their importance for both atmospheric
chemistry and the plant's adaptation to stress, it is necessary to analyze how the surface
processes might change the composition of terpenes reaching the free atmosphere.
The aim of this work was to determine whether terpenes can be found on the epicuticles of
Scots pine (*Pinus sylvestris* L.) and, if so, to compare the spectra of the terpenes with those
found in shoot emissions. To our knowledge this is the first time shoot terpene emissions are
compared with terpenes on needle surfaces of the same tree.
**2    Materials and methods**
We measured shoot-level emissions of pine seedlings at a remote outdoor location in Central
Finland (Hyytiälä, 61°51'N, 24°17'E). The subsequent needle surface wax analysis was
performed in the laboratory of the Finnish Meteorological Institute in Helsinki.
The plant material consisted of four grafted Scots pine seedlings, grown for five years in an
outdoor plant nursery field. Grafted material was selected to reduce variation in the emissions,
since it is well known that the spectrum of terpene emissions depends, among other factors,
on the genetic background (Bäck et al., 2012). The height of the seedlings was 1.5–2 m. The
trees were transplanted in 15 l plastic pots in May 2013. The plants were kept outdoors in
light shade and were well watered. Emission measurements were done during the first days of
August. Scots pine terpene emissions have an annual and a diurnal pattern (Hakola et al.,
2006, Holzke et al., 2006, Ruuskanen et al., 2007, Aalto et al., 2015); the measurement period



was selected to capture sesquiterpene emissions that peak in the summer (Hakola et al., 2006,
Tarvainen et al., 2005).
We aimed to measure the terpene emissions of each seedling once in similar environmental
conditions close to noon and to take three needle samples from each seedling for subsequent
wax analysis.

## 7    2.1    Terpene emissions at shoot level

We measured terpene emissions from the seedlings with a dynamic chamber. The chamber
consisted of a steel frame, coated with PTFE tubing, and a FEP bag supported by the frame
(volume 4.5 l). The chamber was fitted with an inlet and outlet tube made of PTFE. An
external pump, with an active carbon filter and an ozone scrubber, pushed air through the
chamber (2.5 l/min). The chamber system is described in more detail in Hakola et al. (2006).
A healthy mid-crown branch was selected for the emission measurement. Before
measurement, the tip of the branch (approximately 30 cm) was gently fitted in the frame. The
measured section included needles grown in 2013 and 2012. The growth of the new needles
was not quite complete at the time of measurement. The FEP bag was then pulled over the
frame, the pump was started and the system was left to stabilize for 30 minutes to minimize
the effect of emissions induced by handling.
A sample flow was then directed through adsorbent tubes (Tenax-TA and Carbopack-B)
attached to the inlet and outlet tubes with a stainless steel T piece. The resin filling of the tube
adsorbs terpenes, which can later be desorbed and analyzed. Handheld pumps were used to
pull the sample through the tube (70 ml/min). The sampling time was 30 minutes, after which
the chamber was removed. The air temperature inside and the PAR above the chamber were
measured during chamber closure with thermistors (Philips KTY 80/110) and quantum sensor
(LI-190SZ), respectively. During the 60-minute closure, the temperature inside the chamber
increased by 1.5–3 degrees Centigrade. The same chamber was used to measure all the
seedlings. To minimize the effect of changing light conditions, the measurements were done
between 10 AM and 1 PM, which allowed us to measure one tree per day. Each tree was
measured once. After emission measurement and needle sampling (as described below), the
measured shoot was cut and weighed for fresh and dry mass. A 10 % subsample was taken
and weighed separately. For this subsample, we measured needle dimensions (length, width





and thickness) and calculated needle area according to Tirén (1927). This needle area was
then used to estimate the needle area for the shoot using the respective dry weights of the
subsample and main sample.
The contents of the adsorbent tubes were analyzed at the Finnish Meteorological Institute
with a thermal desorber (Perkin-Elmer TurboMatrix 650 ATD) connected to a gas
chromatograph – mass spectrometer (Perkin-Elmer Clarus 600) with HP-1 column (60 m, i.d.
0.25 mm). The detection limits were 0.04 ng/sample for camphene, 0.10–0.15 ng/sample for
α-pinene, β-pinene and carene, 0.20–0.42 ng/sample for sabinene, limonene, 1,8-cineol,
bornylacetate and β-caryophyllene and 0.55–0.64 ng/sample for sesquiterpenes. The measured
compounds were identified using authentic standards and NIST library.
The observed emission rate (E, $\mu g/m^2/h$) was calculated based on the two concentrations of
each compound as
$$E = \frac{(C_2 - C_1)}{A} F \qquad (1)$$
Where $C_2$ is the concentration in the outlet air ($\mu g/m^3$), $C_1$ is the concentration in the inlet air
($\mu g/m^3$), F is the flow rate into the enclosure ($m^3/h$) and A is the needle area of the measured
shoot ($m^2$). From E, we obtained the spectra of emitted compounds (% of total emissions).

## 2.2   Terpenes in the epicuticular waxes

To detect the presence of terpenes associated to the epicuticular surfaces, we collected the
waxy material from the needle surfaces for subsequent terpene analysis.
After each emission measurement, we darkened the measured tree for 30 minutes to close the
stomata and minimize stomatal terpene emission and then took needle samples (20 needle
pairs) in darkness for the wax analysis. The needles were immediately stored in a liquid
nitrogen dry shipper until analysis (two weeks later).
We collected the epicuticular wax layer by dipping each needle pair in 5 ml dichloromethane
for 15 seconds. The dipping time was optimized in a preliminary experiment to remove most
of the wax layer but to keep the solvent from reaching the inside of the needle through
stomata (visual inspection under a stereo microscope). We took special care to use only intact
needles and to not immerse the cut base of the needle in the solvent. This was done to prevent





compounds originating inside the needle from getting into the extract. Dipping the needles
while they were frozen should also minimize the extraction of compounds from inside the
needle. After wax extraction, the needles were weighed for fresh and dry mass and measured
for their dimensions (width, length and thickness). From these dimensions, needle surface
area was approximated according to Tirén (1927).
The obtained extract was evaporated to 1 ml volume with pure nitrogen gas. The reduced
extract was then analyzed with a gas chromatograph (Agilent 6890N) with a mass
spectrometric detector (Agilent 5973) to identify terpenes. A JandW DB-5MS column (30 m,
i.d. 0.25 mm) and a 5 m pre-column (Agilent FS) were used for the chromatography. The
limits of detection were estimated from the standard deviations of blank samples and were
0.15-0.30 ng/sample for p-cymene, bornyl acetate and iso-longifolene, 0.48–0.72 ng/sample
for α-pinene, camphene, myrcene, 1,8-cineol and longicyclene and 1.55–2.29 ng/sample for
β-pinene, 3-carene and β-caryophyllene. The analysis method is described in more detail in
Vestenius et al. (2011). The compounds to be identified were not predetermined, and hence
we did not have calibration standards for all of them. Some of the compounds were therefore
identified and quantified only tentatively, using the reference from another compound. After
the analysis the extract was left to evaporate, and the solid wax residue left in the vial was
weighed (Mettler AT2000).
For an estimation of the terpenes lost during the evaporation, we performed a separate
evaporation test, letting known concentrations of selected terpenes evaporate as described
above. The test gave no indication of any significant loss of terpenes associated with the
method.

## 3   Results

The weather conditions during the experiment were slightly variable. The first two days
(measuring emissions from trees 1 and 2) were relatively warm (+19–21 °C during the
measurements) but partly cloudy. The last two days were sunny and warm, especially the last
day (+21–24 °C). This deserves notice, since the amount of terpenes emitted by a plant is
affected by temperature, irradiation and humidity that on one hand regulate the biosynthetic
processes that produce BVOCs and on the other hand affect volatilization and diffusion rates
(Lerdau and Gray, 2003, Niinemets et al., 2004, Tarvainen et al., 2005).



## 3.1 Terpenes in shoot emissions

The shoot emissions were clearly dominated by monoterpenes (96–98 % of total terpene emissions, Fig. 2). Sesquiterpenes amounted to 0–2 % of total emissions. The compounds found in each group and the variation in their emissions are presented in detail in Appendix B and Fig. 2.

The most abundant monoterpenes were α-pinene (36–58 % of total emissions), myrcene (13–36 %) and carene (12–18 %). The emitted sesquiterpenes included α-humulene (0–1 % of total emissions), aromadendrene (0–0.5 %) and longicyclene (0–0.8 %). None of the identified sesquiterpenes was detected in the emissions of all four pine seedlings, and one seedling showed no sesquiterpene emission. In addition, 1,8-cineol was observed in the emissions, as was a small percentage of bornyl acetate.

## 3.2 Terpenes in epicuticular waxes

The wax yield from the pine needles was 0.0066–0.0114 g/ g DW (average 0.0075 g/g) or 0.43–1.23 g/m$^2$ of needles (average 0.76 g/m$^2$). As for the shoot emissions, the epicuticular wax extracts were dominated by monoterpenes (76–93 % of total terpene amount). The proportion of sesquiterpenes, however, was notably higher than in emissions: 5–21 %. Taking into account the six unidentified sesquiterpenes for which we did not have standards for (described below), the proportion of total sesquiterpenes in the waxes rises to 7–50 % (average 34 %).

The results for different compounds were highly variable also in the wax analysis (Appendix B). The most abundant monoterpenes in the waxes were α-pinene (10–57 % of total), carene (11–26 %) and limonene (2–40 %) (Fig. 2). For sesquiterpenes, the highest amounts were measured for β-caryophyllene (4–16 % of total), iso-longifolene (0–9 %) and humulene (0.5–3 %). Of the sesquiterpenes seen in shoot emissions, only α-humulene was found in the surface waxes. Iso-longifolene was found in the waxes but not in emissions. In addition to the pre-selected compounds (with standards available), we detected six unidentified sesquiterpenes, some in relatively high proportions. This group is likely to include cadinene,





cubebene and murolene. Also 1,8-cineol was found in the waxes, but in much smaller
proportion than in emissions.
**4   Discussion**
**4.1   The terpene spectra in emissions and pine epicuticular waxes**
The composition of the emitted pine shoot terpenes measured in this study is generally in the
range observed by others (Bäck et al., 2012, Hakola et al., 2006, Holzke et al., 2006,
Tarvainen et al., 2005), allowing for the natural variation in BVOC emission and the
differences in methodology. The pine seedlings in our study emitted more than twice as much
α-pinene than carene, thus representing the pinene or intermediate chemotype described in
Bäck et al. (2012). The fact that the pine seedlings were grafted (genetically identical
canopies) is likely to have reduced the variation in the results. Grafted seedlings have the
advantage of providing, at least in theory, identical replicates that should only show variation
caused by differences either in the environmental conditions or life histories (mechanical
injuries, insect attacks and similar). Nevertheless, notable variation in the emissions was
observed, underlining the importance of the effects of varying conditions and life history
experienced by individual trees on their terpene emissions.
The amount of terpenes found in the epicuticular waxes is the equivalent to 4–84 hours of the
measured emissions fort he same compound (per $m^2$ of needle surface), depending on the
compound. For example, it would take the shoot on average 14 h to emit the amount of α-
pinene that was present on the needle surfaces. For myrcene the time would be 9 hours, for
carene 24 hours and for limonene 84 hours. For most sesquiterpenes this comparison cannot
be done, because they were found in either only emissions or only epicuticular waxes, but for
α-humulene the equivalent time would be 34 hours.
The short exposure to the solvent and the fact that the stomata were virtually closed means
that any BVOCs found in the extract were most likely not a result of stomatal emissions but
rather compounds that had been associated to the epicuticle. In studies with extracts from
crushed needles, the proportion of mono- and sesquiterpenes has been found to be in the same
range as observed here for both emissions and epicuticular waxes. For example Manninen et
al., (2002) reported a mean total monoterpene ratio of 67 % for a Scots pine provenance from
central Finland and listed α-pinene and carene as the major monoterpenes in the needles. In



our study, these two were among the main compounds in both emissions and waxes. Achotegui-Castells et al. (2013) reported camphene, α-pinene, β-pinene, β-caryophyllene and germacrene D as the most abundant terpenes in Scots pine needles. Limonene, in our study the third most abundant compound in waxes, was notably less abundant in whole needles (Achotegui-Castells et al., 2013, Manninen et al., 2002). On the other hand camphene was relatively more abundant both in the whole-needle extracts (Achotegui-Castells et al., 2013, Manninen et al., 2002) and in the emissions in our study than in the needle waxes. This is a strong indication that the solvent used in our study did not reach the needle interior during the procedure.

In the epicuticular waxes, we observed six unidentified sesquiterpenes, some in relatively high proportions. Although this group is likely to include cadinene, cubebene and murolene, the exact identification and quantification of these compounds would require a more detailed study. Naturally, the possible role of these compounds in the emissions remains unknown, but their existence in the waxes suggests that the production of sesquiterpenes in Scots pine deserves more attention.

It is interesting to note that despite the large variation there is some indication that the most water-soluble compound in our study, 1,8-cineol, (Appendix A) was relatively more abundant in the emissions, while the compounds with a large $K_{OW}$ (more likely to partition into the lipid than the water phase), like α-humulene, β-caryophyllene and iso-longifolene, were relatively more abundant in the surface waxes. This finding is in line with the results of Welke et al. (1998), who found the cuticular matrix to be a much stronger sink for limonene than for isoprene from air. The compounds with the highest reactivities towards ozone (α-humulene and β-caryophyllene; Appendix A) were more abundant in the epicuticular waxes than emissions. Since the inlet air used in our experiment was scrubbed of ozone, the result is not due to $O_3$-VOC reactions inside the chamber.

## 4.2   The fate of terpenes on leaf surfaces

In theory, there are three mechanisms for the terpenes produced by a needle to end up on the needle surface. The first one is (dry) redeposition after emission. This route is more likely for the less volatile terpenes like longicyclene and p-cymene (Appendix A). The most lipophilic terpenes, such as β-caryophyllene and α-humulene, are also the most reactive ones. Although



they are more likely to bind into or onto the lipophilic wax layer, they are also most unlikely
to survive in the air phase long enough for redeposition to happen (Atkinson and Arey, 2003).
The observed spectra, with β-caryophyllene observed in the pine epicuticular waxes but not in
the shoot emissions and with α-humulene being relatively more abundant in the waxes than
the emissions, are an indication that this route can be considered of minor importance. This
conclusion is supported by Cape et al. (2009), who observed that α-pinene did not dissolve or
adsorb into a wax layer to enhance $O_3$ removal. Another way for the emitted compounds to
bind onto the epicuticular waxes after emission into the air is absorption or adsorption into or
onto the layer of water on the surface.
The second option is transport in the aqueous layer extending from the outer needle surface
through the stoma all the way into the substomatal cavity, as suggested by Burkhardt et al.
(2012). The effectiveness of this route depends on the existence of such continuous water
film, and also on the water-solubility and diffusion capabilities in water of the compound in
question. Because of their low water solubility, it has often been assumed that the reactions of
terpenes in the aqueous phase do not contribute significantly to the total reactions. Wang et al.
(2012) however propose that the reactions of biogenic unsaturated hydrocarbons happening
on wet surfaces, like those of plants growing in nature, can have a significant effect on ozone
deposition. In this work, we cannot differentiate between compounds that were in or on the
epicuticular waxes from those that may have been bound in the surface water. The most
water-soluble of the detected compounds was 1,8-cineol, which was present in greater
proportion in shoot emissions than epicuticular waxes. It is then possible that some of the 1,8-
cineol emitted from the shoot is redeposited onto the surface.
The third alternative is direct transport from the production sites inside the cells through the
plant cuticle. In xerophytic plants, such as conifers, the cuticle has a strongly layered
structure. The insoluble lipid cutin is partly embedded as intracuticular wax under the cuticle
proper, not as an even layer but forming legs towards the epidermal cell wall (Evert, 2007,
Fig. 1). The production of surface waxes takes place in epidermal cells during the first few
weeks and months of needle growth (Kinnunen et al, 1998), and they are transported via
microchannels or diffusion to the surface (Evert, 2007). Despite some reports of terpene
emissions through the cuticle (e.g. Guenther et al, 1991), this route is usually considered
negligible for terpene emissions (Niinemets and Reichstein, 2003) because of the
considerably slower diffusion rate of terpenes within the cuticle than in air or water. It does


not, however, contradict the notion that terpenes might be transported into the epicuticulum
and accumulate there. Theoretically, this mode of transport would be more effective for the
most lipophilic compounds like α-humulene and β-caryophyllene (Kirsch et al., 1997,
Appendix A). Indeed, these compounds were found in greater proportion in the waxes than in
the emissions, suggesting that this may be an important pathway for lipophilic terpenoids.
**4.3  Implications for gas-phase chemistry**
Once in the gas phase, plant-emitted terpenes can react in various ways. They can undergo
photolysis or react with hydroxyl or nitrate radicals or ozone (Atkinson and Arey, 2003). The
relative importance of the different reaction pathways depends on atmospheric conditions,
time of day and the compound in question. Ozone reactions target double bonds in the terpene
molecule (Atkinson and Arey, 2003). The most $O_3$-reactive compounds have two or three of
these double bonds in their structure (Atkinson and Arey, 2003, Appendix A).
The available reaction rate coefficients for $O_3$-BVOC reactions are almost exclusively for the
gas phase (Appendix A). This makes rate calculations of reactions happening on wet plant
surfaces challenging. There is evidence that the reaction rates of terpenes on solid and liquid
surfaces can be faster than in the gas phase (Shen et al., 2013, Enami et al., 2010), but because
of the almost unlimited variation in surface properties and aqueous solutions found in nature,
a single coefficient will never cover all circumstances. For a rough estimate of the $O_3$
scavenging potential of the terpenes we found on the needle surfaces, we calculated their
"reaction time" or how many hours worth of non-stomatal deposition of $O_3$ each compound
could react with, assuming there were no other sinks, as

$$Time = \frac{n_{terp}}{dep_{O3}}$$  (2)
Where Time is the reaction time (h), $n_{terp}$ is the amount of the terpene in question ($\mu g/m^2$) and
$dep_{O3}$ ($\mu g/m^2/h$) is non-stomatal deposition towards the shoot.

Similarly to Fares et al. (2012), we assumed that each molecule of any terpene can react with
one molecule of $O_3$, even though some terpenes have more than one double bond available





while others have none. Assuming a total $O_3$ deposition of 30 ng/m$^2$/s towards the shoot with
40 % non-stomatal deposition (realistic values for Scots pine in the area in the summer as
reported by Altimir et al., 2006), the terpenes present on the surfaces could in theory react
with 5 hours of nonstomatal $O_3$ deposition.
Although simple, our calculation shows that the terpenes found in needle surface waxes could
act as a significant $O_3$ sink. The extent to which this actually happens depends on two factors:
how much of the atmospheric ozone reaches the terpenes within a given time, and how fast
the terpene supply is replenished. The fact that reactive terpenes were present on the needle
surfaces indicates that under the conditions of this study, the terpene supply is renewed at
least at a rate comparable to the amount of $O_3$ reaching the storage site. Assessing these
factors would present an interesting question for future research.
**Appendix A: Physicochemical properties of BVOCs (at 25 °C)**
The values for molecular mass (M),water solubility, Henry's law constant (H), saturated
vapor pressure (VP) and octanol-water partition coefficient (KOW) from Copolovici and
Niinemets (2005) unless otherwise marked. Reaction rate constants from Shu and Atkinson
(1995) unless otherwise marked.

|  | M | Water sol. | H | VP | log[KOW] | Rate constants for gas phase reactions, cm3/molec s | | |
|---|---|---|---|---|---|---|---|---|
|  | g/mol | mol/m$^3$ | Pa m$^3$/mol | Pa | mol/mol | OH | O$_3$ | NO$_3$ |
| p-cymene | 134.2 | 0.21 | 935 | 197** | 4.1 | 8.5x10$^{-12}$ *** | | |
| α-pinene | 136.2 | 0.0411 | 13590 | 558** | 4.66 | 5.4x10$^{-11}$ * | 8.7x10$^{-17}$ * | 6.1x10$^{-12}$ * |
| β-pinene | 136.2 | 0.0592 | 6826 | 404** | 4.42 | 5.7x10$^{-11}$ *** | 1.2x10$^{-17}$ *** | |
| camphene | 136.2 | 0.0419 | 3238 | 136* | 4.56 | 5.7x10$^{-11}$ *** | 1.1x10$^{-17}$ *** | |
| Δ3-carene | 136.2 | 0.0214 | 13640 * | 292* | 4.61 | 8.8x10$^{-11}$ * | 3.7x10$^{-17}$ * | 9.1x10$^{-12}$ * |
| limonene | 136.2 | 0.0886 | 2850 | 253* | 4.49 | 1.5x10$^{-10}$ *** | 4.4x10$^{-16}$ *** | |
| myrcene | 136.2 | 0.0421 | 6300 | 265* | 4.34 | 1.9x10$^{-10}$ *** | 4.4x10$^{-16}$ *** | |
| 1,8-cineole | 154.2 | 19.1 | 13.27 | 253* | 2.61 | 2.3x10$^{-11}$ *** | | |
| bornyl acetate | 196.3 | 0.118 ** | 44.3 *** | 30.4*** | 3.86 ** | 7.7x10$^{-12}$ *** | | |
| longicyclene | 204.4 | 0.966 *** | 2422 *** | 11.5*** | 5.60 *** | 9.4x10$^{-12}$ *** | | |
| iso-longifolene | 204.4 | 0.375 *** | 25939 *** | 6.4*** | 6.12 *** | 9.6x10$^{-11}$ *** | 1.1x10$^{-17}$ *** | |
| β-caryophyllene | 204.4 | 0.245 *** | 69914 *** | 4.2*** | 6.30 *** | 2.0x10$^{-10}$ | 1.2x10$^{-14}$ | 1.9x10$^{-11}$ |
| aromadendrene | 204.4 | 0.345 *** | 29688 *** | 5.3*** | 6.13 *** | 6.2x10$^{-11}$ *** | 1.2x10$^{-17}$ *** | |
| α-humulene | 204.4 | 0.0683 *** | 165160 *** | 2.0*** | 6.95 *** | 2.9x10$^{-10}$ | 1.2x10$^{-14}$ | 3.5x10$^{-11}$ |

Water sol., H, VP, log[KOW]: *)Niinemets and Reichstein (2002) **)Niinemets and Reichstein (2003)
***)ChemSpider. Reaction rate constants: *)Rinne et al., (2007) ***)ChemSpider.





3  **Appendix B: BVOCs in shoot emissions and suface waxes**

| | | a-pinene | myrcene | carene | limonene | b-pinene | camphene | p-cymene | monoterpenes total | a-humulene | aromadendrene | longicyclene | iso-longifolene | b-caryophyllene | unknown1 | unknown2 | unknown3 | unknown4 | unknown5 | unknown6 | sesquiterpenes total | 1,8-cineol | bornylacetate | others total | TOTAL |
|---|---|---|---|---|---|---|---|---|---|---|---|---|---|---|---|---|---|---|---|---|---|---|---|---|---|
| Emissions, µg/m²/h | Tree 1 | 6.8 | 6.7 | 2.6 | 0.0 | 1.6 | 0.8 | 0.0 | **18.4** | 0.0 | 0.0 | 0.0 | 0.0 | 0.0 | | | | | | | **0.0** | 0.4 | 0.0 | **0.4** | **18.8** |
| | Tree 2 | 12.4 | 4.9 | 3.4 | 0.7 | 2.0 | 3.1 | 0.0 | **26.5** | 0.2 | 0.0 | 0.2 | 0.0 | 0.0 | | | | | | | **0.5** | 0.4 | 0.1 | **0.5** | **27.5** |
| | Tree 3 | 13.2 | 3.6 | 4.0 | 0.0 | 1.0 | 0.7 | 0.0 | **22.5** | 0.3 | 0.0 | 0.0 | 0.0 | 0.0 | | | | | | | **0.3** | 0.1 | 0.0 | **0.1** | **22.9** |
| | Tree 4 | 20.0 | 6.0 | 7.9 | 5.3 | 3.3 | 1.2 | 0.4 | **44.1** | 0.2 | 0.2 | 0.0 | 0.0 | 0.0 | | | | | | | **0.4** | 0.4 | 0.1 | **0.5** | **45.0** |
| | Min | 6.8 | 3.6 | 2.6 | 0.0 | 1.0 | 0.7 | 0.0 | **18.4** | 0.0 | 0.0 | 0.0 | 0.0 | 0.0 | | | | | | | **0.0** | 0.1 | 0.0 | **0.1** | **18.8** |
| | Max | 20.0 | 6.7 | 7.9 | 5.3 | 3.3 | 3.1 | 0.4 | **44.1** | 0.3 | 0.2 | 0.2 | 0.0 | 0.0 | | | | | | | **0.5** | 0.4 | 0.1 | **0.5** | **45.0** |
| | Mean | 13.1 | 5.3 | 4.4 | 1.5 | 2.0 | 1.4 | 0.1 | **27.9** | 0.2 | 0.1 | 0.1 | 0.0 | 0.0 | | | | | | | **0.3** | 0.4 | 0.0 | **0.4** | **28.6** |
| | SD | 5.4 | 1.4 | 2.4 | 2.5 | 1.0 | 1.2 | 0.2 | **11.3** | 0.1 | 0.1 | 0.1 | 0.0 | 0.0 | | | | | | | **0.2** | 0.2 | 0.0 | **0.2** | **11.5** |
| In waxes, µg/m² | Tree 1 s 1 | 62.9 | 1.0 | 29.7 | 2.3 | 0.0 | 2.9 | 0.0 | **98.8** | 1.1 | 0.0 | 0.0 | 0.2 | 9.2 | 0.2 | 3.5 | 21.4 | 8.5 | 0.0 | 2.7 | **46.8** | 0.2 | 3.5 | **3.7** | **149** |
| | Tree 1 s 2 | 408 | 3.6 | 147 | 44.2 | 21.1 | 10.0 | 9.1 | **642** | 11.8 | 0.0 | 0.0 | 39.8 | 83.2 | 13.0 | 24.2 | 158 | 104 | 2.6 | 26.8 | **464** | 3.3 | 10.1 | **13.3** | **1120** |
| | Tree 1 s 3 | 20.1 | 2.5 | 9.9 | 6.2 | 0.0 | 0.0 | 1.5 | **38.8** | 0.4 | 0.0 | 0.0 | 0.3 | 3.0 | 0.1 | 2.3 | 16.8 | 8.5 | 0.0 | 0.8 | **32.1** | 0.0 | 1.1 | **1.1** | **72.1** |
| | Tree 2 s 1 | 120 | 9.3 | 39.8 | 20.7 | 0.0 | 3.9 | 0.5 | **194** | 5.2 | 0.0 | 0.0 | 7.7 | 39.8 | 5.3 | 12.8 | 62.5 | 43.4 | 1.3 | 17.3 | **195** | 1.1 | 3.8 | **4.9** | **394** |
| | Tree 2 s 2 | 59.0 | 5.8 | 32.2 | 18.2 | 11.9 | 4.4 | 0.5 | **132** | 4.8 | 0.0 | 0.0 | 1.3 | 25.0 | 2.7 | 5.0 | 29.4 | 10.8 | 1.3 | 14.7 | **94.9** | 1.1 | 10.2 | **11.2** | **238** |
| | Tree 2 s 3 | 213 | 372 | 463 | 856 | 83.9 | 0.0 | 0.0 | **1988** | 14.5 | 0.0 | 0.0 | 0.0 | 112 | 0.0 | 3.1 | 18.7 | 5.7 | 0.0 | 1.9 | **156** | 18.7 | 3.9 | **22.6** | **2166** |
| | Tree 3 s 1 | 152 | 21.6 | 71.9 | 61.2 | 8.4 | 6.9 | 1.1 | **324** | 4.4 | 0.0 | 0.0 | 3.4 | 36.0 | 2.2 | 8.1 | 48.1 | 17.7 | 0.6 | 11.4 | **132** | 2.0 | 5.6 | **7.6** | **463** |
| | Tree 3 s 2 | 76.3 | 11.6 | 25.8 | 38.7 | 9.8 | 7.0 | 1.9 | **171** | 2.2 | 0.0 | 0.0 | 4.5 | 14.3 | 0.6 | 6.8 | 49.9 | 15.0 | 1.1 | 1.9 | **96.4** | 1.3 | 5.4 | **6.7** | **274** |
| | Tree 3 s 3 | 305 | 22.4 | 132 | 62.4 | 12.5 | 11.0 | 1.5 | **547** | 11.2 | 0.0 | 0.0 | 25.0 | 83.2 | 12.0 | 21.8 | 108 | 104 | 4.6 | 48.9 | **376** | 2.6 | 12.4 | **15.0** | **938** |
| | Tree 4 s 1 | 421 | 7.0 | 81.6 | 14.5 | 20.2 | 18.1 | 3.1 | **565** | 8.6 | 0.0 | 0.0 | 66.0 | 69.7 | 10.9 | 19.4 | 159 | 64.3 | 3.8 | 12.3 | **414** | 3.3 | 21.4 | **24.7** | **1004** |
| | Tree 4 s 2 | 207 | 101 | 152 | 355 | 39.0 | 10.0 | 2.6 | **867** | 4.8 | 0.0 | 0.0 | 7.6 | 37.0 | 1.6 | 8.6 | 70.3 | 10.7 | 1.5 | 0.0 | **142** | 8.4 | 5.9 | **14.3** | **1023** |
| | Tree 4 s 3 | 82.5 | 21.6 | 69.3 | 60.2 | 11.4 | 2.8 | 0.0 | **248** | 4.7 | 0.0 | 0.0 | 1.8 | 34.4 | 2.5 | 8.0 | 30.9 | 12.7 | 1.3 | 18.9 | **115** | 1.7 | 1.5 | **3.1** | **366** |
| | Min | 20.1 | 1.0 | 9.9 | 2.3 | 0.0 | 0.0 | 0.0 | **38.8** | 0.4 | 0.0 | 0.0 | 0.0 | 3.0 | 0.0 | 2.3 | 16.8 | 5.7 | 0.0 | 0.0 | **32.1** | 0.0 | 1.1 | **1.1** | **72.1** |
| | Max | 421 | 372 | 463 | 856 | 83.9 | 18.1 | 9.1 | **1988** | 14.5 | 0.0 | 0.0 | 66.0 | 112 | 13.0 | 24.2 | 159 | 104 | 4.6 | 48.9 | **464** | 18.7 | 21.4 | **24.7** | **2166** |
| | Mean | 177 | 48.3 | 105 | 128 | 18.2 | 6.4 | 1.7 | **485** | 6.1 | 0.0 | 0.0 | 13.1 | 45.5 | 4.3 | 10.3 | 64.4 | 30.3 | 1.5 | 13.2 | **189** | 3.6 | 7.1 | **10.7** | **684** |
| | SD | 137 | 106 | 123 | 248 | 23.4 | 5.2 | 2.6 | **537** | 4.4 | 0.0 | 0.0 | 20.6 | 33.9 | 4.9 | 7.6 | 51.1 | 31.4 | 1.5 | 14.2 | **146** | 5.2 | 5.7 | **7.6** | **598** |

| | | a-pinene | myrcene | carene | limonene | b-pinene | camphene | p-cymene | monoterpenes total | a-humulene | aromadendrene | longicyclene | iso-longifolene | b-caryophyllene | sesquiterpenes total | 1,8-cineol | bornylacetate | others total | TOTAL |
|---|---|---|---|---|---|---|---|---|---|---|---|---|---|---|---|---|---|---|---|
| Emissions, % of total | Tree 1 | 35.9 | 35.7 | 13.6 | 0.0 | 8.4 | 4.0 | 0.0 | **97.7** | 0.0 | 0.0 | 0.0 | 0.0 | 0.0 | **0.0** | 2.3 | 0.0 | **2.3** | **100.0** |
| | Tree 2 | 44.9 | 17.8 | 12.3 | 2.7 | 7.2 | 11.4 | 0.2 | **96.4** | 0.8 | 0.0 | 0.8 | 0.0 | 0.0 | **1.7** | 1.6 | 0.3 | **1.9** | **100.0** |
| | Tree 3 | 57.5 | 15.8 | 17.4 | 0.1 | 4.5 | 3.0 | 0.0 | **98.3** | 1.2 | 0.0 | 0.0 | 0.0 | 0.0 | **1.2** | 0.5 | 0.0 | **0.5** | **100.0** |
| | Tree 4 | 44.4 | 13.4 | 17.5 | 11.8 | 7.3 | 2.7 | 0.8 | **97.9** | 0.5 | 0.5 | 0.0 | 0.0 | 0.0 | **0.9** | 1.0 | 0.2 | **1.2** | **100.0** |
| | Min | 35.9 | 13.4 | 12.3 | 0.0 | 4.5 | 2.7 | 0.0 | **96.4** | 0.0 | 0.0 | 0.0 | 0.0 | 0.0 | **0.0** | 0.5 | 0.0 | **0.5** | **100.0** |
| | Max | 57.5 | 35.7 | 17.5 | 11.8 | 8.4 | 11.4 | 0.8 | **98.3** | 1.2 | 0.5 | 0.8 | 0.0 | 0.0 | **1.7** | 2.3 | 0.3 | **2.3** | **100.0** |
| | Mean | 45.7 | 20.7 | 15.2 | 3.7 | 6.8 | 5.3 | 0.2 | **97.6** | 0.6 | 0.1 | 0.2 | 0.0 | 0.0 | **1.0** | 1.4 | 0.1 | **1.5** | **100.0** |
| | SD | 8.9 | 10.2 | 2.6 | 5.6 | 1.7 | 4.1 | 0.4 | **0.8** | 0.5 | 0.2 | 0.4 | 0.0 | 0.0 | **0.7** | 0.8 | 0.1 | **0.8** | **0.0** |
| In waxes, % of total | Tree 1 s 1 | 55.7 | 0.8 | 26.3 | 2.0 | 0.0 | 2.6 | 0.0 | **87.4** | 0.9 | 0.0 | 0.0 | 0.2 | 8.1 | **9.2** | 0.2 | 3.1 | **3.3** | **100.0** |
| | Tree 1 s 2 | 51.6 | 0.5 | 18.5 | 5.6 | 2.7 | 1.3 | 1.2 | **81.3** | 1.5 | 0.0 | 0.0 | 5.0 | 10.5 | **17.1** | 0.4 | 1.3 | **1.7** | **100.0** |
| | Tree 1 s 3 | 46.1 | 5.8 | 22.6 | 14.2 | 0.0 | 0.0 | 0.3 | **89.0** | 0.8 | 0.0 | 0.0 | 0.8 | 6.9 | **8.5** | 0.0 | 2.6 | **2.6** | **100.0** |
| | Tree 2 s 1 | 47.7 | 3.7 | 15.8 | 8.2 | 0.0 | 1.5 | 0.2 | **77.1** | 2.1 | 0.0 | 0.0 | 3.1 | 15.8 | **20.9** | 0.4 | 1.5 | **1.9** | **100.0** |
| | Tree 2 s 2 | 33.9 | 3.3 | 18.4 | 10.5 | 6.8 | 2.5 | 0.3 | **75.7** | 2.7 | 0.0 | 0.0 | 0.7 | 14.4 | **17.8** | 0.6 | 5.8 | **6.4** | **100.0** |
| | Tree 2 s 3 | 10.0 | 17.4 | 21.7 | 40.0 | 3.9 | 0.0 | 0.0 | **93.0** | 0.7 | 0.0 | 0.0 | 0.0 | 5.2 | **5.9** | 0.9 | 0.2 | **1.1** | **100.0** |
| | Tree 3 s 1 | 40.7 | 5.8 | 19.2 | 16.3 | 2.3 | 1.8 | 0.3 | **86.3** | 1.2 | 0.0 | 0.0 | 0.9 | 9.6 | **11.7** | 0.5 | 1.5 | **2.0** | **100.0** |
| | Tree 3 s 2 | 38.4 | 5.8 | 13.0 | 19.5 | 4.9 | 3.5 | 0.9 | **86.1** | 1.1 | 0.0 | 0.0 | 2.2 | 7.2 | **10.6** | 0.6 | 2.7 | **3.4** | **100.0** |
| | Tree 3 s 3 | 44.8 | 3.3 | 19.4 | 9.2 | 1.8 | 1.6 | 0.2 | **80.3** | 1.6 | 0.0 | 0.0 | 3.7 | 12.2 | **17.5** | 0.4 | 1.8 | **2.2** | **100.0** |
| | Tree 4 s 1 | 57.3 | 1.0 | 11.1 | 2.0 | 2.7 | 2.5 | 0.4 | **77.0** | 1.2 | 0.0 | 0.0 | 9.0 | 9.5 | **19.6** | 0.4 | 2.9 | **3.4** | **100.0** |
| | Tree 4 s 2 | 22.2 | 10.9 | 16.4 | 38.1 | 4.2 | 1.1 | 0.3 | **93.2** | 0.5 | 0.0 | 0.0 | 0.8 | 4.0 | **5.3** | 0.9 | 0.6 | **1.5** | **100.0** |
| | Tree 4 s 3 | 28.3 | 7.4 | 23.7 | 20.6 | 3.9 | 1.0 | 0.0 | **84.9** | 1.6 | 0.0 | 0.0 | 0.6 | 11.8 | **14.0** | 0.6 | 0.5 | **1.1** | **100.0** |
| | Min | 10.0 | 0.5 | 11.1 | 2.0 | 0.0 | 0.0 | 0.0 | **75.7** | 0.5 | 0.0 | 0.0 | 0.0 | 4.0 | **5.3** | 0.0 | 0.2 | **1.1** | **100.0** |
| | Max | 57.3 | 17.4 | 26.3 | 40.0 | 6.8 | 3.5 | 1.2 | **93.2** | 2.7 | 0.0 | 0.0 | 9.0 | 15.8 | **20.9** | 0.9 | 5.8 | **6.4** | **100.0** |
| | Mean | 39.7 | 5.5 | 18.8 | 15.5 | 2.8 | 1.6 | 0.3 | **84.3** | 1.3 | 0.0 | 0.0 | 2.2 | 9.6 | **13.2** | 0.5 | 2.0 | **2.5** | **100.0** |
| | SD | 14.1 | 4.8 | 4.4 | 12.6 | 2.1 | 1.0 | 0.4 | **6.0** | 0.6 | 0.0 | 0.0 | 2.6 | 3.6 | **5.4** | 0.3 | 1.5 | **1.5** | **0.0** |



# 1   Acknowledgements

Anni Vanhatalo, Ditte Mogensen, Theo Kurtén and Pontus Roldin are acknowledged for their
valuable help before, during and after the experiment. We thank the Natural Resources
Institute Haapastensyrjä unit for the grafted plant material. The research was supported by the
Academy of Finland Center of Excellence (grant no. 272041), Maj and Tor Nessling
foundation, the Finnish Society of Forest Science and the Doctoral Programme in Sustainable
use of renewable natural resources (AGFOREE). N. Altimir thanks VOCBAS for supporting
the exchange visit where the initial idea for this study was generated.



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



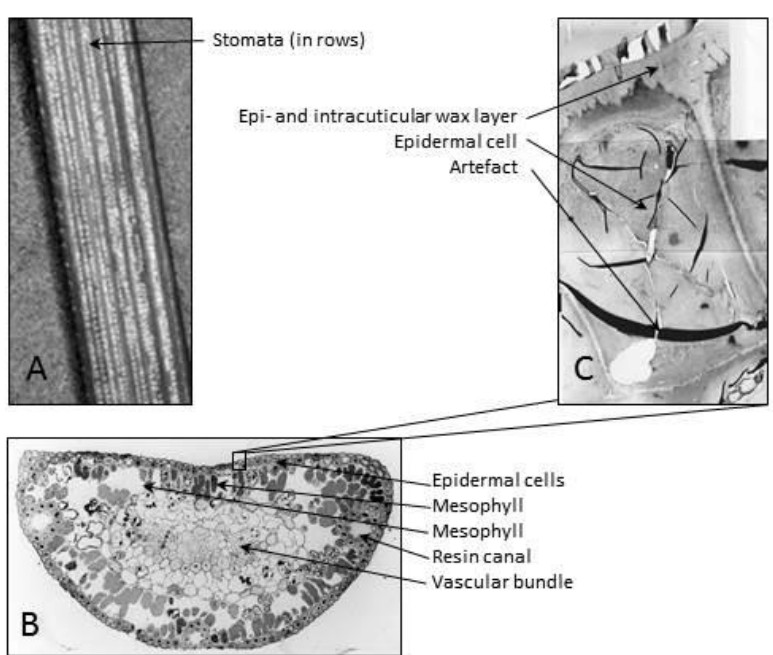

2 Fig. 1. Pine needle structure. A. The abaxial side of the needle with rows of stomata covered

3 with epicuticular waxes. B. Cross-section of a needle. C. An epidermal cell with epicuticular

4 layer.



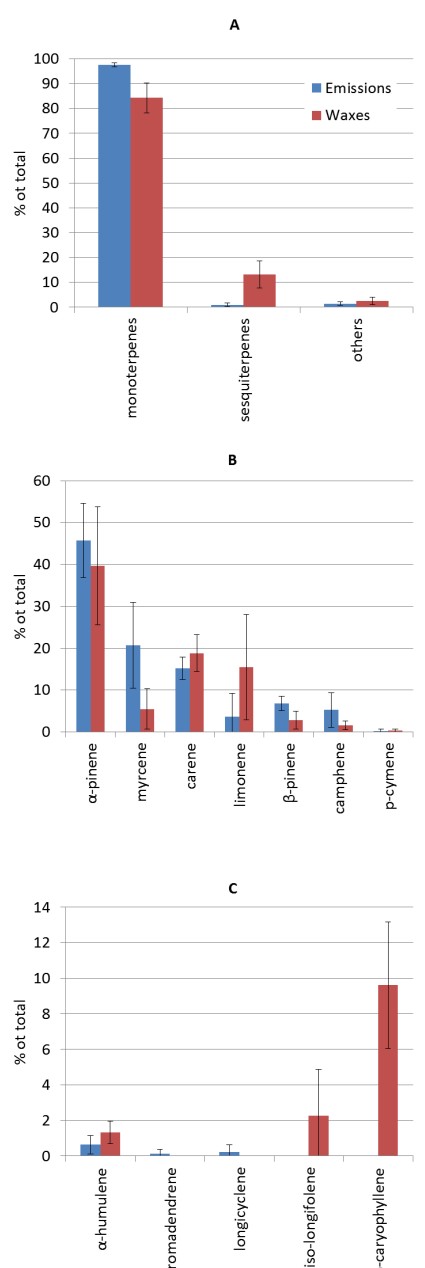

Fig. 2. Relative amounts of terpenes in the pine shoot emissions and needle surface waxes,
average % of total, with standard deviation. A: relative abundancies of each compound group,
B: monoterpenes, C: sesquiterpenes. The unknown sesquiterpenes found in the waxes are not
included.