# Peer review of "Role of needle surface waxes in dynamic exchange of"

_Atmospheric Chemistry and Physics, 2015_

## Referee Comment (RC1) · Anonymous Referee #1 · 16 Feb 2016

General comments: One of the motivations for this study is non-stomatal ozone deposition. Significant non-stomatal ozone fluxes have been frequently observed, but a general explanation has not been given yet. While there are no obvious reaction sites for ozone reaction in leaf surface waxes, dissolved or attached terpenes could react effectively with ozone. This idea has been around for a while, but although it was not entirely supported by first experiments, a thorough characterization of the system is still missing and might enable more successful experiments in the future. From this background, the present contribution adds importantly to the knowledge of the fate of terpenes shortly after synthesis and their possible role in ozone depsition. The authors compare the composition of mono- and sesquiterpenes emitted from pine shoots with their abundance in needle waxes. There are common compounds but also some compounds which appear only in one of the compartments. Alternative ways of transport

are discussed. Although the general message of the manuscript is clear, the presentation of the results doesn't seem appropriate to me. Fig. 1 C shows results for compounds where no detection limits are given (a-humulene, aromadendrene). Especially the part with missing standards for some of the compounds measured remains weak and the high amount of sesquiterpenes claimed (up to 50%) doesn't seem to be sufficiently corroborated. Which of the three mentioned compounds (cadinene, cubebene, murolene) would be most abundant? There were also very high differences between repetitions of the same tree (e.g., Tree 2, myrcene: 9, 6, and 372 $\mu$g m-2; Tree 4, limonene: 15, 355, and 60 $\mu$g m-2). While notable variations between the emissions are mentioned in the discussion, these differences are not discussed. Have similarly large differences been reported before or how could they be explained? Could this be an indication that the solvent was not equally effective?

Specific comments: Due to the indicated artifact and some other unexplained structures, Figure 1C is not very fortunate. It should be possible to find a better series of photographs, or sketch to illustrate the relevant features.

P, 5, L. 21: Were the 'handheld pumps' operated by persons and how could they do this evenly for 30 minutes? If they were machine controlled, why were they handheld?

Technical corrections: P. 2, l. 30: 'or', not 'on'

---

## Referee Comment (RC2) · Anonymous Referee #2 · 21 Mar 2016

General comments: Authors have measured whole shoot-level mono- and sesquiterpene emissions of Scots pine seedlings and analysed the needle surface waxes for the same compounds. The aim of the work was to determine if the same terpenes can be found on the epicuticles as in shoot emissions. This approach is needed to better understand the mechanisms how plant release BVOCs in the atmosphere and if there is a temporal storage of BVOCs on plant surfaces.

Main observations were that shoot emissions and wax extracts were dominated by monoterpenes and the proportion of some sesquiterpenes was higher in the wax extracts than in whole shoot emissions. Authors have discussed about the pathways of mono- and sesquiterpenes to needle cuticle also considering external sources. Their conclusion was the "any BVOCs found in the extract were most likely not a result of stomatal emissions but rather compounds that had been associated to the epicuticle".

[Figure]

However, whole discussion is based on the assumption that needle emissions are the only source of needle epicuticular mono- and sesquiterpenes. External redeposition is mentioned, but other possible external sources are not discussed. These might include e.g. emissions from the bark of studied branches or other branches and stem, but also emission from neighboring plants. Authors should mention these other pathways of needle deposition of BVOCs.

Specific comments:

P 4, L24. The analysis is based only on four seedlings, so crafted shoots representing the same genotype was a good choice.

P5, L12. Air flow in the shoot chambers was rather high. How much this may stimulate monoterpene and sesquiterpene emission from bark?

P 6, L 22-23. Three replicate samples were reported. How they were collected? Was each of those composed of 20 needles or were these 20 needles divided to 3 subsamples?

P10, L9. Redeposition plant's own BVOCs on epicuticular waxes might not be the only pathway. Adsorption of sesquiterpenes on epicuticular wax layer from external plant sources and their emission back to atmosphere is reported (Li & Blande 2015). As Scots pine bark is important monoterpene and sesquiterpene emitter (e.g. Ghirardo et al. 2012, Heijari et al. 2011). There could be a possibility that part of detected sesquiterpenes on epicuticular wax may originate from earlier sesquiterpene emission from bark of the focal plant and neighboring plants and adsorbed on needles?

P 14. Appendix B. Authors should discuss about potential reason for high variation in monoterpene content in replicate samples within each tree. As the same sample has high emission of all common resin monoterpenes (e.g. tree 2 s3 and tree 4 s2), it may suggest e.g. high bark emission from micro cracks near these needles. Together with high sesquiterpene content in some of the needle samples localized biotic stress by

e.g. fungal pathogen or mites might also explain these.

References

Ghirardo et al. (2010) Plant, Cell and Environ. 33, 781–792 Li, T & Blande JD, (2015) Global Change Biology 21, 1993–2004 Heijari et al. (2011) Environ. Exp. Bot. 71, 390–398.

---

## Author Comment (AC1) · 2 May 2016

Dear Referee, We thank you for your thorough work and valuable comments. We have made most of the suggested changes; where this was not possible, a more thorough explanation is given below.

Comments from Referee

General comments: One of the motivations for this study is non-stomatal ozone deposition. Significant non-stomatal ozone fluxes have been frequently observed, but a general explanation has not been given yet. While there are no obvious reaction sites for ozone reaction in leaf surface waxes, dissolved or attached terpenes could react effectively with ozone. This idea has been around for a while, but although it was not entirely supported by first experiments, a thorough characterization of the system is

still missing and might enable more successful experiments in the future. From this background, the present contribution adds importantly to the knowledge of the fate of terpenes shortly after synthesis and their possible role in ozone deposition.

The authors compare the composition of mono- and sesquiterpenes emitted from pine shoots with their abundance in needle waxes. There are common compounds but also some compounds which appear only in one of the compartments. Alternative ways of transport are discussed. Although the general message of the manuscript is clear, the presentation of the results doesn't seem appropriate to me. Fig. 1 C shows results for compounds where no detection limits are given (a-humulene, aromadendrene). Especially the part with missing standards for some of the compounds measured remains weak and the high amount of sesquiterpenes claimed (up to 50%) doesn't seem to be sufficiently corroborated. Which of the three mentioned compounds (cadinene, cubebene, murolene) would be most abundant?

There were also very high differences between repetitions of the same tree (e.g., Tree 2, myrcene: 9, 6, and 372g m-2; Tree 4,limonene: 15, 355, and 60g m-2). While notable variations between the emissions are mentioned in the discussion, these differences are not discussed. Have similarly large differences been reported before or how could they be explained? Could this be an indication that the solvent was not equally effective?

Specific comments: Due to the indicated artifact and some other unexplained structures, Figure 1C is not very fortunate. It should be possible to find a better series of photographs, or sketch to illustrate the relevant features.

P, 5, L. 21: Were the 'handheld pumps' operated by persons and how could they do this evenly for 30 minutes? If they were machine controlled, why were they handheld?

Technical corrections: P. 2, l. 30: 'or', not 'on'

Response to comments

The authors compare the composition of mono- and sesquiterpenes emitted from pine shoots with their abundance in needle waxes. There are common compounds but also some compounds which appear only in one of the compartments. Alternative ways of transport are discussed. Although the general message of the manuscript is clear, the presentation of the results doesn't seem appropriate to me. Fig. 1 C shows results for compounds where no detection limits are given (a-humulene, aromadendrene). Especially the part with missing standards for some of the compounds measured remains weak and the high amount of sesquiterpenes claimed (up to 50%) doesn't seem to be sufficiently corroborated. Which of the three mentioned compounds (cadinene, cubebene, murolene) would be most abundant?

This comment points to Fig 1 C, but judged by the content it is meant to be 2 C, and our response is based on this assumption. There are essentially two points in the comment: 1) We have now added the missing detection limits for a-humulene and aromadendrene. These compounds do not exist in blank samples, but we calculated the blank levels by integrating background noise of the chromatogram. 2) In addition to quantifying compounds known to be emitted from pine shoots, we wanted to search for any indication of possible additional compounds in the waxes. For this reason, we searched the library for candidate compounds for all unidentified large peaks we observed. However, since we did not have the standards, we do not know the actual responses, and an analysis of the possible relative abundances of these compounds is not possible.

There were also very high differences between repetitions of the same tree (e.g., Tree 2, myrcene: 9, 6, and 372g m-2; Tree 4,limonene: 15, 355, and 60g m-2). While notable variations between the emissions are mentioned in the discussion, these differences are not discussed. Have similarly large differences been reported before or how could they be explained? Could this be an indication that the solvent was not equally effective?

This is a very valuable comment. The variation in the terpene content of the epicuticular waxes cannot be explained by variation in wax yield (i.e. solvent effectiveness). Even though there is variation in wax yield (per needle area), this variation does not correspond to the variation observed in the terpenes. We do not know of previous studies with similar methodology, so there is nothing to compare to. It is possible that some of the variation was caused by the sampling procedure. Despite the short sampling time, it is possible that the emissions caused by plucking needles had sufficient time to adsorb onto other needles that were subsequently picked into a sample. Other possible causes of variation include small cracks, insect bites or pathogens, in the bark near some of the needles. E.g. insect bites are known to induce both local and systemic terpene emissions (Heijari et al., 2011). Some of these may well have escaped visual inspection. One very likely source is true natural variation between needles grown in different parts of the branch/canopy. Very little is known on this topic, but since terpene synthesis is light-dependent, it is very likely that there are differences (Juho Aalto, personal communication). The wax yields have been added to Appendix B and the possible causes of variation have been discussed more thoroughly.

Specific comments: Due to the indicated artifact and some other unexplained structures, Figure 1C is not very fortunate. It should be possible to find a better series of photographs, or sketch to illustrate the relevant features.

The photographs collaged to produce 1C are quite old, and we know their quality could be better. However, we feel that the image illustrates an important feature not often discussed in literature (the fact that the epicuticular waxes are actually present not just on the surface but all around the epicuticular cell). Unfortunately it is not possible for us to acquire a better photograph, and a drawing would not be sufficiently credible for this purpose.

P, 5, L. 21: Were the 'handheld pumps' operated by persons and how could they do this evenly for 30 minutes? If they were machine controlled, why were they handheld?

"Handheld" referred actually only to the small size of the pumps; they were batteryoperated. The wording has been changes to "small pumps" to avoid unnecessary confusion.

Technical corrections: P. 2, l. 30: 'or', not 'on' These mistakes have been corrected, thank you for noticing them!

Changes made in the manuscript based on these comments

P2 L 30: "on" corrected to "or" P5 L 21: "Small pumps were used to pull the sample through the tube (70 ml/min)." (instead of "handheld pumps" P6 L7-8: Added: "The detection limits were ... 0.05 ng/sample for $\alpha$-humulene and aromadendrene, ..." P7 L22: Added: "The limits of detection ... were 0.15-0.30 ng/sample for ..., $\alpha$-humulene, aromadendrene..." P9 L26 New paragraph: "The is remarkable variation observed in the terpene content of the epicuticular waxes, and this variation cannot be explained by variation in the amount of extracted wax. Possible natural causes of variation include small cracks, insect bites or pathogens in the bark near some of the needles. E.g. insect bites are known to induce both local and systemic terpene emissions (Heijari et al., 2011). Some of these may well have escaped visual inspection. One feasible source is true natural variation between needles grown in different parts of the branch or canopy, due to the light-delendent nature of terpene synthesis. Very little is known on this topic, but it is very likely that there are notable differences (Juho Aalto, personal communication). Some of the variation, however, may have been caused by the sampling procedure itself. Despite the short sampling time, it is possible that the emissions caused by plucking needles had sufficient time to adsorb onto other needles that were subsequently picked into a sample. " APPENDIX B: Added: Wax yields References: Added: Heijari, J., Blande, J.D. and Holopainen, J.K. Feeding of large pine weevil on Scots pine stem triggers localised bark and systemic shoot emission of volatile organic compounds, Environ Exp Bot, 71, 390-398, 2011.

[revised manuscript text omitted]

---

## Author Comment (AC2) · 2 May 2016

Dear Referee, We thank you for your thorough work and valuable comments. We have made of the suggested changes, as we feel they clearly improved the quality of the manuscript.

Comments from Referee

General comments: Authors have measured whole shoot-level mono- and sesquiter-pene emissions of Scots pine seedlings and analysed the needle surface waxes for the same compounds. The aim of the work was to determine if the same terpenes can be found on the epicuticles as in shoot emissions. This approach is needed to better understand the mechanisms how plant release BVOCs in the atmosphere and if there is a temporal storage of BVOCs on plant surfaces.

Main observations were that shoot emissions and wax extracts were dominated by monoterpenes and the proportion of some sesquiterpenes was higher in the wax extracts than in whole shoot emissions. Authors have discussed about the pathways of mono- and sesquiterpenes to needle cuticle also considering external sources. Their conclusion was the "any BVOCs found in the extract were most likely not a result of stomatal emissions but rather compounds that had been associated to the epicuticle". However, whole discussion is based on the assumption that needle emissions are the only source of needle epicuticular mono- and sesquiterpenes. External redeposition is mentioned, but other possible external sources are not discussed. These might include e.g. emissions from the bark of studied branches or other branches and stem, but also emission from neighboring plants. Authors should mention these other pathways of needle deposition of BVOCs.

Specific comments: P 4, L24. The analysis is based only on four seedlings, so crafted shoots representing he same genotype was a good choice. P5, L12. Air flow in the shoot chambers was rather high. How much this may stimulate monoterpene and sesquiterpene emission from bark? P 6, L 22-23. Three replicate samples were reported. How they were collected? Was each of those composed of 20 needles or were these 20 needles divided to 3 subsamples? P10, L9. Redeposition plant's own BVOCs on epicuticular waxes might not be the only pathway. Adsorption of sesquiterpenes on epicuticular wax layer from external plant sources and their emission back to atmosphere is reported (Li & Blande 2015). As Scots pine bark is important monoterpene and sesquiterpene emitter (e.g. Ghirardoet al. 2012, Heijari et al. 2011). There could be a possibility that part of detected sesquiterpenes on epicuticular wax may originate from earlier sesquiterpene emission from bark of the focal plant and neighboring plants and adsorbed on needles? P 14. Appendix B. Authors should discuss about potential reason for high variation in monoterpene content in replicate samples within each tree. As the same sample has high emission of all common resin monoterpenes (e.g. tree 2 s3 and tree 4 s2), it may suggest e.g. high bark emission from micro cracks near these needles. Together with high sesquiterpene content in some of the needle samples

localized biotic stress by e.g. fungal pathogen or mites might also explain these.

References

Ghirardo et al. (2010) Plant, Cell and Environ. 33, 781–792 Li, T & Blande JD, (2015) Global Change Biology 21, 1993–2004 Heijari et al. (2011) Environ. Exp. Bot. 71, 390–398.

Response to comments

Main observations were that shoot emissions and wax extracts were dominated by monoterpenes and the proportion of some sesquiterpenes was higher in the wax extracts than in whole shoot emissions. Authors have discussed about the pathways of mono- and sesquiterpenes to needle cuticle also considering external sources. Their conclusion was the "any BVOCs found in the extract were most likely not a result of stomatal emissions but rather compounds that had been associated to the epicuticle". However, whole discussion is based on the assumption that needle emissions are the only source of needle epicuticular mono- and sesquiterpenes. External redeposition is mentioned, but other possible external sources are not discussed. These might include e.g. emissions from the bark of studied branches or other branches and stem, but also emission from neighboring plants. Authors should mention these other pathways of needle deposition of BVOCs.

It is true that the possible sources of the redeposited terpenes are not discussed; this is indeed a valuable remark. We have added a mention of the possible pathways in the discussion.

P5, L12. Air flow in the shoot chambers was rather high. How much this may stimulate monoterpene and sesquiterpene emission from bark?

Since the chamber encloses the whole shoot, some of the emissions measured come from the bark/stem of the shoot, not only the needles. This is true of any shoot measurement done with a similar chamber setup. The biomass inside such a chamber is

typically 10-25 % wood material (including bark). The needles are more active terpene emitters than the wood/bark, but there is some evidence of compound-specific variation (Anni Vanhatalo, personal communication). It is also likely that the needles are more susceptible to any air current induced disturbance than the bark.

P 6, L 22-23. Three replicate samples were reported. How they were collected? Was each of those composed of 20 needles or were these 20 needles divided to 3 subsamples?

We took three separate samples of 20 needle pairs each. This information has been added to the text.

P10, L9. Redeposition plant's own BVOCs on epicuticular waxes might not be the only pathway. Adsorption of sesquiterpenes on epicuticular wax layer from external plant sources and their emission back to atmosphere is reported (Li & Blande 2015). As Scots pine bark is important monoterpene and sesquiterpene emitter (e.g. Ghirardoet al. 2012, Heijari et al. 2011). There could be a possibility that part of detected sesquiterpenes on epicuticular wax may originate from earlier sesquiterpene emission from bark of the focal plant and neighboring plants and adsorbed on needles?

It is true that the possible sources of the redeposited terpenes are not discussed; this is indeed a valuable remark. We have added a mention of the possible pathways in the discussion.

P 14. Appendix B. Authors should discuss about potential reason for high variation in monoterpene content in replicate samples within each tree. As the same sample has high emission of all common resin monoterpenes (e.g. tree 2 s3 and tree 4 s2), it may suggest e.g. high bark emission from micro cracks near these needles. Together with high sesquiterpene content in some of the needle samples localized biotic stress by e.g. fungal pathogen or mites might also explain these.

This is a very valuable comment. The variation in the terpene content of the epicuticular waxes cannot be explained by variation in wax yield (i.e. solvent effectiveness). Even though there is variation in wax yield (per needle area), this variation does not correspond to the variation observed in the terpenes. We do not know of previous studies with similar methodology, so there is nothing to compare to. It is possible that some of the variation was caused by the sampling procedure. Despite the short sampling time, it is possible that the emissions caused by plucking needles had sufficient time to adsorb onto other needles that were subsequently picked into a sample. Other possible causes of variation do indeed include small cracks, insect bites or pathogens, in the bark near some of the needles. Some of these may well have escaped visual inspection. One very likely source is true natural variation between needles grown in different parts of the branch/canopy. Very little is known on this topic, but since terpene synthesis is light-dependent, it is very likely that there are differences (Juho Aalto, personal communication). The wax yields have been added to Appendix B and the possible causes of variation have been discussed more thoroughly.

P11 L26: Changed to: "The second option is transport in the aqueous layer . . . This route is naturally only available to terpenes produced by the needle itself, and the effectiveness of the route depends on the existence of such a continuous water film. . ."

APPENDIX B: Added: Wax yields

[revised manuscript text omitted]

---

## Author Response (AR2)

Authors' response to editor

Dear Editor,

Thank you for your thorough work towards the technical quality of the manuscript! We have made all the suggested technical corrections. The list of references did indeed have a few inconsistencies. The list has been double-checked to conform to the Web of Science Journal Title Abbreviations. "Global Biogeochem Cy" is indeed the abbreviation for the journal Global Biogeochemical Cycles.